# Cardiolipin, Non-Bilayer Structures and Mitochondrial Bioenergetics: Relevance to Cardiovascular Disease

**DOI:** 10.3390/cells10071721

**Published:** 2021-07-08

**Authors:** Edward S. Gasanoff, Lev S. Yaguzhinsky, Győző Garab

**Affiliations:** 1STEM Program, Science Department, Chaoyang KaiWen Academy, Beijing 100018, China; 2Bioenergetics Department, Belozersky Research Institute for Physico-Chemical Biology, Lomonosov Moscow State University, 119992 Moscow, Russia; yag@belozersky.msu.ru; 3Moscow Institute of Physics and Technology, 141701 Dolgoprudny, Russia; 4Institute of Cytochemistry and Molecular Pharmacology, 115404 Moscow, Russia; 5Department of Physics, Faculty of Science, University of Ostrava, 71000 Ostrava, Czech Republic; garab.gyozo@brc.hu; 6Biological Research Center, H-6726 Szeged, Hungary

**Keywords:** cardiolipin, non-bilayer structures, electron-transport chain, inner mitochondrial membrane, ATP synthase, cardiovascular disease

## Abstract

The present review is an attempt to conceptualize a contemporary understanding about the roles that cardiolipin, a mitochondrial specific conical phospholipid, and non-bilayer structures, predominantly found in the inner mitochondrial membrane (IMM), play in mitochondrial bioenergetics. This review outlines the link between changes in mitochondrial cardiolipin concentration and changes in mitochondrial bioenergetics, including changes in the IMM curvature and surface area, cristae density and architecture, efficiency of electron transport chain (ETC), interaction of ETC proteins, oligomerization of respiratory complexes, and mitochondrial ATP production. A relationship between cardiolipin decline in IMM and mitochondrial dysfunction leading to various diseases, including cardiovascular diseases, is thoroughly presented. Particular attention is paid to the targeting of cardiolipin by Szeto–Schiller tetrapeptides, which leads to rejuvenation of important mitochondrial activities in dysfunctional and aging mitochondria. The role of cardiolipin in triggering non-bilayer structures and the functional roles of non-bilayer structures in energy-converting membranes are reviewed. The latest studies on non-bilayer structures induced by cobra venom peptides are examined in model and mitochondrial membranes, including studies on how non-bilayer structures modulate mitochondrial activities. A mechanism by which non-bilayer compartments are formed in the apex of cristae and by which non-bilayer compartments facilitate ATP synthase dimerization and ATP production is also presented.

## 1. Mitochondrial Dysfunction Is Linked to Health Deterioration

### 1.1. Organization of Mitochondrial Membranes Supports High Energy Demand of Organisms 

Mitochondrion is a subcellular biochemical energy powerhouse, which generates ATP molecules via the process of oxidative phosphorylation. ATP in a sufficient quantity is required for supporting all biochemical and biophysical reactions needed for the adequate maintenance of all physiological processes in a body. ATP is also needed to support apoptosis and autophagy, processes leading to the programmed cell death. Mitochondria are made of two membranes, the outer mitochondrial membrane (OMM) and the inner mitochondrial membrane (IMM). The space between the OMM and IMM is called the intermembrane space. The mitochondrial matrix is the space separated by the IMM from the intermembrane space. The IMM is composed of numerous invaginations called cristae and the space inside cristae is called the intracristae space, which is also part of the intermembrane space. The respiratory protein complexes, CI, CII, CIII, and CIV, which make up the electron transport chain (ETC), and CV (the F_0_F_1_-ATP synthase), are embedded in the cristae of the IMM. The transfer of protons, H^+^, from the matrix into the intracristae space, a process coupled with the redox reactions that sequentially transfer electrons through the respiratory protein complexes CI to CIV, generates a proton gradient across the IMM, which is needed for driving the production of ATP by the F_0_F_1_-ATP synthase. In cells of tissues with a high energy demand, such as heart muscles and skeletal musculature, there is an increased number of large and elongated mitochondria with numerous and tightly stacked cristae membranes, which take up most of the mitochondrial volume [1].

### 1.2. Decline in ATP Production Is Linked to Cardiovascular and Other Diseases and Aging

Proper functioning of important organs and tissues such as heart, brain, kidneys, retina, skeletal musculature, the network of nerve cells, and immune system cells, requires large amounts of ATP. Abnormal decline in cellular ATP production impairs repair of injured cells, neurotransmission, muscle contraction, blood circulation, etc., which leads to inflammation, ischemia, heart failure and other diseases and may cause premature death of cells and an entire organism. A decline in ATP production is also linked to aging [2,3]. With age, the number of mitochondria in cells decreases, and mitochondria become smaller in size and more round in shape, and the number of cristae in the IMM decreases and cristae become shorter in length [4,5,6,7]. A decrease in overall surface area of cristae results in the decreased activity of the respiratory protein complexes, especially CI and CIV [8,9,10], which ultimately accounts for the decreased production of ATP [11]. It has been observed in the elderly and in patients with cardiovascular and other diseases that decline in ATP production impairs pathways of programmed cell death [12], resulting in an increased rate of morbidity and mortality.

## 2. Cardiolipin and Mitochondrial Bioenergetics

### 2.1. Conical Shape of Cardiolipin Promotes Formation of Cristae

The exclusive presence of cardiolipin in the energy-generating membranes of mitochondria points to a profound role of cardiolipin in bioenergetics. In mitochondria, cardiolipin accounts for about 2 mol% of phospholipids in the OMM and for about 20 mol% of phospholipids in the IMM [13]. Cardiolipin is made of four acyl hydrocarbon chains, which are linked through the glycerol’s ester bonds with two phosphate groups in the cardiolipin anionic head [14]. Fatty acids of cardiolipin are tissue specific [1,13]. Cardiolipin has a reverse wedge molecular shape, also called conical shape, in which the transverse area of the cardiolipin’s anionic head is smaller than the transverse area of the four alkyl chains. The conical shape of cardiolipin can be increased via the interaction of its phosphate groups with basic proteins like Szeto–Schiller tetrapeptides made of cationic and hydrophobic amino acid residues [1] (Figure 1A). Due to its increased conical shape, cardiolipin exerts lateral pressure in a bilayer membrane that induces negative curvature and causes the inner monolayer of IMM to bend to form cristae curvatures [15], as shown in Figure 1B. Moreover, due to its conical shape, cardiolipin has a high propensity to form non-bilayer structures of which are presumed to be mainly developed in the apex of cristae. 

### 2.2. Cardiolipin Increases Efficiency of Electron-Transport Chain and ATP Synthase Activity

The surface area of the IMM is more than four times larger than that of the OMM [16]. Shortage in cardiolipin results in decreases in length and number of cristae, and in the overall surface area of IMM, which, in turn, reduces the ATP production [17]. It should be noted that apart from supporting the cristae architecture, cardiolipin is needed for the overall stability and functional activity of respiratory protein complexes in the IMM. It has been shown that cardiolipin binds to all proteins of the respiratory complex [18] and the presence of cardiolipin between proteins of CI, CII, CIII, CIV, and the ATP synthase is required for the optimal functional activities of all proteins in the IMM [19,20]. Cardiolipin is concentrated at the apex of cristae where it not only helps in stabilizing cristae curvature but also supports ATP synthase dimerization to optimize the production of ATP [21,22,23,24]. Cardiolipin acts as a ‘glue’ to stabilize the proteins of the ETC, which is essential in oligomerization of respiratory complexes into respirasomes [25]. Electrostatic interaction between cardiolipin and proteins of respiratory complexes decreases the distance between redox partners, which facilitates electron transfer [26,27,28]. It is important to note that cardiolipin also electrostatically interacts with the cationic cytochrome *c* to bring it closer to the ETC, and by this means, accelerates the rate of electron transfer from CIII to CIV [29]. In addition, the anionic phosphate groups of cardiolipin are capable of entrapping protons for the ATP synthase dimers, which are located predominantly in the cristae tips [30]. Cardiolipin is also crucial for the stabilization of the multi-subunit machinery of mitochondrial contact site and cristae organizing system (MICOS) [31]. MICOS is important for forming the structural basis of cristae junctions [31], which are tubular structures demarcating the cristae from the inner boundary membrane. It has been proposed that the transmembrane proton gradient across the IMM could be generated via the flip-flopping of protonated cardiolipin or via transient channels in the IMM formed at the protein/cardiolipin interface [32]. It has also been hypothesized that anionic cardiolipin can serve as a proton shuttler [33]. This hypothesis was recently further developed into a concept in which a proton-shuttling may generate a proton potential gradient along the membrane surface of the IMM to power the rotation of ATP synthase complex [32]. This concept is somewhat consistent with an early view in which cardiolipin acts as a proton trap by sequestering protons from the ETC via anionic phosphate groups of cardiolipin with subsequent release of protons to the F_0_ subunit of the ATP synthase [33]. In the very recent series of publications, it has been demonstrated that non-bilayer phospholipid structures, which are made predominantly of cardiolipin in the IMM, facilitate ATP synthase activity through a molecular mechanism, which is not yet fully understood [34,35,36,37]. Although there are still many details remain enigmatic in molecular mechanisms concerning the role of cardiolipin in supporting the activities of the ETC and the ATP synthesis, it becomes increasingly clear that cardiolipin is an indispensable molecule in the ETC and ATP synthesis of IMM, which is required for proper physiological structure and functioning of mitochondria.

## 3. Cardiolipin and Cardiovascular Disease

### 3.1. Higher Cardiolipin Concentration Boosts Higher Mitochondrial Density and Tighter Density of Cristae 

Mitochondria in cardiac muscle cells are the most resistant to aging [38], which is presumably due to the higher concentration of cardiolipin in the IMM of mitochondria in cardiomyocytes, where they constitute about 25 mol% of the total phospholipids content; in comparison in the IMMs of other cells, they represent about 20 mol% of the total phospholipids [13]. Cardiomyocytes also have the highest mitochondrial density totaling to 30% of the total intracellular volume [39] and the highest density of cristae in the IMM [1]. Higher concentration of cardiolipin and tighter density of cristae in the IMM along with higher density of mitochondria in cardiomyocytes all help cardiac muscle cells to meet high demand for energy needed for the adequate heart functioning. A subtle mitochondrial dysfunction leading to a slight decline in mitochondrial ATP production in cardiomyocytes is harmful to the cardiovascular health and may lead to ischemia and heart failure [40]. 

### 3.2. Decline in Cardiolipin Leads to Progression of Cardiovascular Disease 

A decline in the concentration of cardiolipin in the IMM leads to mitochondrial dysfunction in several ways. Interaction of cardiolipin with all proteins of electron transport chain (ETC) is essential for proper functioning of ETC [18,19,20,26,27,28,40]. Oxidation of cardiolipin by reactive oxygen species triggers inactivation of complexes I, III, and IV [40,41]. Cardiolipin is also needed for proper assembly, structural stability and efficient functioning of ETC supercomplexes [40]. Deficiency in cardiolipin leads to destabilization of complexes III and IV in the supercomplexes in yeast cells [42], dissociation of complex IV from the supercomplexes and decline in levels of I/III supercomplexes in lymphoblasts of patients with Barth syndrome [43]. In addition, cardiolipin interacts with transporter proteins including phosphate carrier [44], pyruvate carrier [45] and ADP-ATP carrier [46]. Thus, deficiency in cardiolipin impairs activities of these carrier proteins, ultimately leading to a decreased production of ATP [40]. Finally, aberrant decrease of cardiolipin in the IMM leads to other forms of mitochondrial dysfunction such as defective protein import and mitophagy with the subsequent progression to cardiovascular disease [40]. 

The most obvious connection between mitochondrial cardiolipin deficiency and cardiovascular disease is demonstrated in Barth syndrome, a genetic disorder linked to over 160 mutations in X chromosome [47,48,49]. The major clinical manifestation of Barth syndrome is cardiomyopathy [50]. The other symptoms of Barth syndrome are growth retardation, myopathy, and neutropenia [51]. Biochemical ‘fingerprints’ of Barth syndrome include mitochondrial cardiolipin deficiency in cardiac muscle cells [52], increased levels of monolysocardiolipin, altered alkyl chains of cardiolipin [53,54], and tafazzin deficiency [49]. It is the tafazzin deficiency that triggers pathological changes resulting in decreased cardiolipin, increased monolysocardiolipin, and in cardiolipin species with altered alkyl chains, all of which cause cardiomyopathy leading to a cardiovascular disease [40]. 

Several other cases of mitochondrial dysfunction associated with the decreased levels of cardiolipin in the IMM that trigger abnormalities in the respiratory chain, decline in ATP synthesis, and loss in the structural integrity of mitochondria, leading to the cardiovascular disease pathogenesis, are caused by the genetic disorders, such as Senders syndrome and dilated cardiomyopathy with ataxia, both of which inhibit cardiolipin biosynthesis [55,56]. Other cardiovascular diseases, which are associated with the decline in cardiolipin level in the IMM, include ischemia–reperfusion injury and heart failure. These cardiovascular diseases also highlight the role of a decreased level of cardiolipin in an aberrant mitochondrial biogenesis and morphology leading to apoptosis and mitophagy with implications in cardiovascular disease [55].

## 4. Non-Bilayer Structures and Possible Implications in Cardiovascular Disease

### 4.1. Non-Bilayer Lipids in the Energy Transducing Membranes

Conically shaped non-bilayer lipids, the cardiolipin, phosphatidylethanolamine (PE), and the monogalactosyl diacylglycerol (MGDG) constitute about half of the total lipid contents of the IMMs and the thylakoid membranes, the energy-transducing membranes of mitochondria and of oxygenic photosynthetic organisms, respectively. Conically shaped lipids lend high non-bilayer propensity to the bulk lipid mixtures of these membranes, which thus impose negative curvature stress on the lamellar membranes and may trigger formation of non-bilayer structures [57]. Phosphatidic acid (PA), which is also found in small quantity in mitochondrial membranes [58,59], acquires conical shape at low pH and in the presence of divalent cations [60], but its tendency to form non-bilayer structures is less expressed than that of cardiolipin or PE [36]. Contrary to PE, which is abundant in many cellular and subcellular membranes [61], cardiolipin is found exclusively in membranes of mitochondria [58]. It accounts for about 2% of phospholipids in the OMM, which exists strictly as a bilayer, and for ~20% in the IMM [58,60], in which polymorphic bilayer to non-bilayer transitions have recently been reported [36]. As to PE, its presence is high both in the OMM and IMM; it accounts for about 25% of phospholipids in both membranes [58,60]. As no non-bilayer structures have been reported in the OMM, one can conclude that cardiolipin has higher propensity to form non-bilayer structures in mitochondrial membranes than PE. As opposed to the neutral and less conically shaped PE, it seems that cardiolipin’s anionic polar head and its high propensity to form non-bilayer structures are the important factors which help cardiolipin in assembling a functionally active and stable structure of the respiratory chain supercomplexes in the IMM. This assumption is supported by the report in which depletion of cardiolipin, but not of PE, has been shown to destabilize the respiratory chain supercomplexes [62]. This is also indirectly supported by observations in model membranes mimicking the phospholipid composition in the IMM: a decrease in pH induced a significant increase in the population of non-bilayer structures in membranes lacking PE but induced only a slight increase in non-bilayer structures in membranes lacking cardiolipin [36, unpublished data]. 

### 4.2. Role of Non-Bilayer Structures in the Energy Transducing Membranes

The role of non-bilayer structures in the energy transducing membranes and in biological membranes, in general, is not well understood. In early studies on lipid polymorphism, non-bilayer lipids were reported to induce frustrated states in the bilayer membrane [63]. Concerning the role of non-bilayer lipids in the bilayer, it was emphasized that non-bilayer lipids generate curvature elastic energy, which could trigger the formation of non-bilayer structures, but only locally and transiently; and thus, no functional importance was assigned to non-bilayer structures [64]. In the bilayer membrane, non-bilayer lipids have been proposed to exert high lateral packing pressure on membrane proteins and, by this means, to keep them in functional states [65]. 

According to an alternative, non-conflicting hypothesis, the amount of non-bilayer lipids relative to the total lipid content of a membrane, i.e., the non-bilayer propensity of a lipid mixture, self-regulates the lipid-to-protein ratio of membranes [66]. This hypothesis is based on two premises: (i) it has earlier been shown that membrane-intrinsic proteins can force non-bilayer lipids to enter the bilayer [67,68]; and (ii) in the absence of a sufficiently high protein concentration, and exposing protein-free membrane areas to water, intermediate interlamellar structures are formed, which then segregate from the bilayer and assemble into (a) non-bilayer structure(s) [69]. (For a molecular dynamic model of spontaneous stalk formation of thylakoid lipids, see [70]). The Janus phase of lipid mixtures with non-bilayer propensity, i.e., being capable of entering the membrane but ready to segregate and leave the bilayer, has been then proposed to have a common role in self-regulating the lipid-to-protein ratio in energy-transducing membranes of mitochondria and chloroplasts [66,71]. Both the IMMs and the thylakoid membranes are very densely packed with proteins, and a ‘dilution’ of these membranes would most certainly perturb the assembly of the protein supercomplexes and their cooperative interactions. 

Extensive studies on lipid polymorphism and dynamic changes between lamellar and non-bilayer structures in thylakoid membranes promoted development of the dynamic exchange model (DEM) for the lipid phases in the light-energy converting membranes. According to this model, bilayer and non-bilayer phases co-exist as a dynamic equilibrium between the different lipid phases [71,72,73,74]. This is most clearly indicated by reversible co-solute-, temperature- and pH-dependent changes in the lipid-phase behavior of thylakoid membranes [73,75]. The shifts in equilibrium may represent states of various physiological activities in these light-energy converting membranes. Recent experimental data have demonstrated that lipid-phase transitions from bilayer to non-bilayer structures affect the membrane permeability and energization of thylakoid membranes [76], and modulate the activity of violaxanthin de-epoxidase (VDE), a water-soluble enzyme, which participates in photoprotection of the photosynthetic machinery in thylakoids [75]. These observations further support the notion that the bilayer to non-bilayer transitions could be a normal physiological occurrence in the functionally active energy transducing membranes.

The concept that polymorphic lipid phase transitions may regulate physiological activities in energy transducing/converting membranes reconciles well with the recent findings that formation of cardiolipin-containing non-bilayer structures, triggered by changes in the temperature, pH or by the action of membrane-active peptides, promote the activity of ATP synthase in bovine heart mitochondria [34,35,36,37]. These findings open new pharmacological avenues for novel methods in treatment of cardiovascular disease. In the following sections of this review article, we discuss new opportunities in treatment of cardiovascular disease by using synthetic and native peptides as potential pharmaceuticals that target cardiolipin in the IMM to reverse mitochondrial dysfunction and rejuvenate mitochondrial bioenergetics.

## 5. Targeting Cardiolipin in the IMM

### 5.1. Szeto-Schiller Tetrapeptides Penetrate Cell Barriers

In order to elucidate molecular mechanisms that explain the role of cardiolipin in regulation of mitochondrial bioenergetics, the cell-penetrating and cardiolipin-targeting Szeto–Schiller (SS) tetrapeptides have been synthesized [77,78]. The effects of two SS peptides, SS-20 and SS-31, which target mitochondria, have been extensively studied in cells, tissues, and animals [1,77,78]. Both tetrapeptides have amino acid sequences with alternating aromatic and basic amino acid residues. In SS-20 peptide, arginine and lysine residues alternate with two phenylalanine residues, H-Phe-D-Arg-Phe-Lys-NH_2_, while in SS-31 peptide, arginine and lysine residues alternate with dimethyl-tyrosine and phenylalanine residues, H-D-Arg-Dmt-Lys-Phe-NH_2_ (Dmt = dimethyltyrosine) [1]. Both SS peptides exhibit the high cell membrane permeability and both SS peptides can penetrate cell barriers with tight junctions, including the blood-brain barrier [79,80]. A mechanism behind the high cell permeability of SS peptides is unclear. It was suggested that electrons in π orbitals in aromatic rings of phenylalanine and dimethyl-tyrosine may shield positive charge of arginine and lysine residues via the cation-π electron interaction to offer ‘stealth’ properties to SS peptides. This may explain a mechanism by which these peptides evade electrostatic repulsion from the choline group of phosphatidylcholines in the outer leaflets of cellular membranes [1].

### 5.2. Szeto-Schiller Tetrapeptides Target Cardiolipin

Once SS peptides are inside the cells, they target mitochondria and penetrate through the OMM and then selectively interact with cardiolipin on the IMM via the force of electrostatic attraction [81,82]. Upon binding to cardiolipin on the IMM, SS peptides can increase oxygen consumption and ATP production [81,82,83] by increasing the efficiency of electron transfer and P/O coupling. This, in turn, reduces electron leak and decreases the accumulation of reactive oxygen species from the ETC [82]. Electron transfer from CIII to CIV via cytochrome *c* is the rate-limiting step in the ETC. It is assumed that electrostatic interaction of cardiolipin with cytochrome *c* keeps cytochrome *c* close to the respiratory protein complexes, which thus facilitates electron transfer [1]. However, it has also been hypothesized that hydrophobic interaction of cytochrome *c* with cardiolipin may disturb the heme environment to make it less conducive for electron transfer. Contrary to this hypothesis, addition of SS peptides to samples of mitochondria did increase oxygen consumption after direct reduction of cytochrome *c* with N,N,N,N′-tetramethyl-1,4-phenylenediamine/ascorbate which strongly suggests that cardiolipin-bound SS peptides can penetrate deep into the cytochrome *c* heme environment to facilitate electron transfer in the crucial rate-limiting step in the ETC [82,83].

### 5.3. Szeto–Schiller Tetrapeptides Activate a Broad Range of Rejuvenation Reactions

Overall, extensive studies of the SS peptides’ actions on mitochondrial activities and their related physiological effects in cells, tissues, and bodies revealed that SS peptides can induce a broad range of physiological and pharmacological reactions, including rejuvenation of oxidative phosphorylation and decreasing the reactive oxygen species production, inhibition of cardiolipin peroxidation, remodeling of mitochondrial cristae structure in aged mice, upregulation of enzymes required for cardiolipin biosynthesis and remodeling, restoration of mitochondrial bioenergetics and dynamics, restoration of cellular structure and function during aging, prevention of cell death and inflammation, and boosting body’s natural ability to heal itself [1]. A detailed mechanism by which SS peptides can induce such a broad range of powerful rejuvenating physiological reactions is unclear. However, it has been suggested that binding of SS peptides to cardiolipin on the IMM may increase phospholipid packing in such a way to induce tighter cristae curvature [1] to make it more conducive for higher efficiency of the ETC to generate more energy, thus enabling restoration of mitochondrial cristae architecture in aged mice (Figure 2). Indeed, restoration of mitochondrial cristae architecture empowers the increased ATP production to provide enough energy needed for a broad range of rejuvenating reactions in cells and a body.

## 6. Mode of Cardiolipin Packing in Model Phospholipid Membranes

### 6.1. Coexistence of Bilayer and Non-Bilayer Phases in Membranes Containing Cardiolipin

Application of physical methods, such as ^31^P-NMR in multilamellar dispersions [36,64,84,85,86,87], EPR of spin probes in oriented lipid films [86,87,88,89], and ^1^H-NMR of unilamellar liposomes in potassium ferricyanide solution [36,85,86,87,88,90,91], made it possible to attain a reliable information about the mode of phospholipid packing in model membranes containing cardiolipin. It was determined that model membranes made of ‘lamellar’ phospholipids, such as phosphatidylcholine, and containing from 5 to 20 mole% cardiolipin exist in physiological solution as a lamellar bilayer phase [64,85,91]. Addition of calcium ions at a concentration of 2.5 mM to solution of sonicated phosphatidylcholine liposomes containing from 5 to 20 mole% cardiolipin induced an increase in turbidity of liposomal samples studied by spectrophotometry at 325 nm. An increase in turbidity was attributed to the aggregation of liposomes [84,91,92]. ^31^P-NMR signals from multilamellar dispersions of phosphatidylcholine containing 20 mol% cardiolipin in 5 mM CaCl_2_ solution revealed the co-existence of two phospholipid phases: inverted hexagonal (H_II_) lipid phase and lamellar bilayer phase [93]. When multilamellar dispersions of phosphatidylcholine containing from 2.5 to 25 mol% cardiolipin were treated with cobra venom protein cardiotoxins, which similarly to SS peptides have basic and aromatic amino acid residues in their sequences, ^31^P-NMR spectroscopy of phospholipids in multilamellar dispersions revealed that phospholipids co-existed in three different lipid phases: (1) non-bilayer structures with immobilized phospholipids, (2) non-bilayer isotropically oriented phospholipids with high molecular mobility, and (3) a lamellar bilayer phase [34,36,84,85,86]. ^1^H-NMR analysis in potassium ferricyanide solution of unilamellar liposomes with phospholipid composition equivalent to the abovementioned multilamellar dispersions analyzed by ^31^P-NMR and treated with the same cardiotoxins, showed that non-bilayer packed phospholipids included not only cardiolipin, but also phosphatidylcholine [36,84,85,86,87,88]. To elucidate the molecular mechanism by which cardiotoxins trigger formation of non-bilayer structures that include both cardiolipin and phosphatidylcholine [36,84,85], additional physical and computational methods were employed to study the interaction of cardiotoxins with cardiolipin-containing membranes. ^2^H-NMR spectra of water bound on membrane surface were recorded to investigate the cardiotoxin-induced dehydration of phospholipid polar heads [85,86]. The method of the EPR of spin probes in oriented lipid films was used to monitor cardiotoxin-induced changes in the spatial orientation of phospholipids’ long molecular axis [86,87,88,89,90]. The technique of hydrophobic probe phosphorescence deactivation was utilized to study intermembrane contacts between neighboring liposomes promoted by cardiotoxin binding to cardiolipin [86,91,94]. Differential scanning calorimetry of dimiristoyl and dipalmitoyl based saturated phospholipids was employed to study cardiotoxin-induced intermembrane exchange of phospholipids between neighboring liposomes [86,91,95]. The method of molecular dynamics was used to simulate atomic details in the initial binding of cardiotoxins to the surface of membrane made of cardiolipin and phosphatidylcholine [36,88]. The docking program, Autodock-Vina, was used to simulate docking interactions of cardiotoxins with cardiolipin and phosphatidylcholine molecules when cardiotoxins are already submerged into the membrane at the interface between phospholipid polar heads and hydrophobic alkyl tails (Figure 3) [33,34,35,36,37,86,88]. 

### 6.2. Cardiotoxin Induced Intermembrane Junction Is Promoted by Non-Bilayer Packed Cardiolipin

Extensive analysis of a multitude of experimental and computational data obtained by using the abovementioned experimental and in silico modeling approaches allowed to develop a tentative converging molecular mechanism by which cardiotoxins interact with the cardiolipin-containing membranes [36,37]. This converging molecular mechanism was also extended to explain molecular details about cardiotoxin interaction with the outer mitochondrial membrane [37]. Analysis of the results from the ^2^H-NMR studies of the cardiotoxin-triggered changes in the dynamics of water bound on membrane surface revealed an important step in the initial interaction of cardiotoxins with a membrane, which is the dehydration of membrane surface by cardiotoxins [85]. It appears that cardiotoxin CTII from *Naja oxiana* venom initially binds via R36 and K12 residues to cardiolipin polar heads on the surface of membrane made of phosphatidylcholine plus 5 mol% cardiolipin. K23 and other charged and polar amino acid residues of cardiotoxin bind to phosphate groups of both cardiolipin and phosphatidylcholine to establish short-range ionic, ion-dipole and hydrogen bonds with the polar groups of cardiolipin and phosphatidylcholine at the interface between phospholipid polar heads and hydrophobic alkyl tails. At the same time, L9 and other hydrophobic amino acid residues of cardiotoxin drive interaction of cardiotoxin with alkyl chains of phospholipids [37]. Several basic residues, K5, K18, K35, and K44, of cardiotoxin do not submerge into the membrane and remain exposed to solvent. These basic residues attract anionic polar heads of phospholipids on a membrane of neighboring liposome to initially dehydrate membrane surface of neighboring liposome, leading to the formation of intermembrane junction with cardiotoxin in its center (Figure 4). 

The intermembrane junction, a contact point between aggregated liposomes, has a non-bilayer phospholipid packing structure. Cardiolipin molecules, which have a reverse wedge molecular shape, are located in intermembrane junction, in areas of high surface curvature [37]. It should be noted that basic and polar amino acid residues that bind to polar head groups of phospholipids are highly conserved across the cardiotoxins of cobra species [37]. This may imply that cardiotoxins share a converging mechanism by which cardiotoxins interact with membranes made of cardiolipin and phosphatidylcholine [37]. It has been shown by thermodynamic stability studies that intermembrane junctions have a life span between 0.1 to 1.0 s [96]. The cardiotoxin-triggered intermembrane junction is a starting point in one of the processes, which include membrane aggregation, intermembrane exchange of lipids, membrane fusion, and translocation of cardiotoxin across the membrane [84,85,91,95,96]. In phosphatidylcholine membranes containing 20 mol% cardiolipin, cardiotoxins translocate easily across membranes with the help of non-bilayer intermembrane junctions [84,85,95]. This is particularly true in a system of multilamellar dispersions, phospholipid composition of which is similar to that of the IMM [37,86,88]. The closely located lamellas in a battery of phospholipid membranes in multilamellar dispersions strongly resemble the structure of tightly packed cristae in the IMM [84]. It is possible that SS peptides, which similarly to cardiotoxins have alternating basic and aromatic amino acid residues, can translocate across tightly packed cristae membranes via the mechanism of non-bilayer intermembrane junctions that could be conducive to the increased efficiency of the ETC, leading to a higher production of ATP.

## 7. A Link between Non-Bilayer Structures and Mitochondrial Bioenergetics

### 7.1. High Non-Bilayer Propensity of Cardiolipin Plays an Important Role in Reshaping Cristae Architecture

Mitochondria have always been considered as the highly dynamic organelles burdened with responsibility to support an array of key metabolic and regulatory reactions. On the level of mitochondrial membranes, the dynamic exchanges between the OMM and IMM have been known for decades. Application of electron microscopy has revealed a great diversity in the cristae architecture adapted for each type of cells, tissues, metabolic conditions, energy demands, and the states of health and disease [97,98]. At the same time, cristae were considered for a long time to be the entities with the conserved and static structure. However, the latest advances in super-resolution techniques have led to the discovery showing that cristae are highly dynamic and independent bioenergetic units with the ability to remodel in a timescale of seconds in response to the changes in energy demands and physiological states of cells and tissues [97,98,99,100]. Extensive studies of the cristae structure by Frey and Mannella, in which they used three-dimensional electron tomography, helped to understand that cristae are not simply extended invaginations of the IMM but are tubular-like membranous ‘sub-organelles’ connected to the inner boundary membrane by the slit-like structures called crista junctions [101,102]. Application of stimulated emission depletion (STED) super-resolution microscopy along with fluorescence recovery after photobleaching and single-particle tracking, made it possible to realize that cristae and cristae junctions undergo extremely dynamic changes linked to the formation and disappearance of cristae or remodeling of cristae membranes all of which taking place in a timeframe of few seconds or less [98,100,103]. Cristae dynamics and local membrane remodeling affects the membrane potential at the level of individual cristae which causes significant impact on oxidative phosphorylation, Ca^2+^ homeostasis, and apoptosis [97,99]. It should be noted that imbalance in cristae and cristae junction dynamics triggers development of diseases [99,104,105]. Cristae-membrane remodeling and formation of cristae junctions are regulated by the complex of the mitochondrial contact site and cristae organizing system (MICOS), optic atrophy type 1 protein (OPA1), ATP synthase, and the lipid microenvironment predominantly composed of cardiolipin [97,100,105,106]. It should be noted that ‘non-bilayer’ propensity of cardiolipin along with MICOS, OPA1, and ATP synthase helps regulating the curvature of the cristae membranes and cristae junctions [99,104,106]. Subunits of the MICOS complex (MIC60 and MIC10) located at crista junctions and surrounded by cardiolipin, control membrane-bending activity [97]. The cardiolipin bound OPA1 regulates cristae width, while short and long forms of OPA1, also bound to cardiolipin, keep cristae junctions tight. ATP synthase dimers surrounded primarily by cardiolipin along with the lipid microenvironment in the inner leaflet of the cristae membrane, composed predominantly of cardiolipin, define positive membrane curvature at the cristae tip [97]. Decline in cardiolipin concentration significantly changes the cristae architecture and the landscape of OXPHOS complexes [107], leading to cardiovascular and neurodegenerative disorders and cancer [98,108]. In the cardiac muscle cells, the loss of mitochondrial cardiolipin, which is induced by tafazzin mutations causal to Barth syndrome in humans, results in development of abnormal swollen cristae arranged as concentric stacks or highly interconnected cristae [109,110], leading to significant defects in skeletal and heart musculature [109].

It should be noted that remodeling of cristae architecture, mediated by cardiolipin along with MICOS, OPA1, and the ATP synthases, is a dynamic process that constantly involves cycles of mitochondrial membrane fusion and fission [111,112,113]. Cardiolipin is a major player supporting mitochondrial fusion. Disruptions in the mitochondrial fusion dynamics, which are triggered by decreased levels of cardiolipin, are caused by aging and human diseases, including cardiovascular diseases [109,111]. In the course of over three decades of research studies on membrane fusion in model membrane systems that mimic phospholipid composition of the IMM, which were conducted by two of the three authors of this review paper, it was revealed that the membrane fusion is driven by the bilayer to non-bilayer polymorphic transitions mediated by cardiolipin in the model IMM systems [84,85,86,87,88,89,91,94,95]. The non-bilayer phospholipid structures in the model IMM systems were recorded by ^31^P-NMR spectroscopy on a timescale 10^−2^ to 10^−4^ s. This finding allows one to conclude that the high propensity of cardiolipin to form non-bilayer structures in the IMM in a timescale of 10^−2^ to 10^−4^ s is likely a key factor that supports the processes behind the remodeling of cristae architecture taking place in a timescale of seconds.

### 7.2. Non-Bilayer Packed Cardiolipin Is Immobilized by Binding to F_0_ Subunits of ATP Synthase 

The phospholipid polymorphic phase transitions in mitochondrial samples were observed for the first time more than four decades ago by ^31^P-NMR spectroscopy [114]. ^31^P-NMR spectra from the aqueous dispersion of the rat liver mitochondrial phosphatidylethanolamine revealed a bilayer-to-hexagonal (H_II_) phase transition in the 10–37 °C temperature range [114]. Aqueous dispersion of phosphatidylcholine from rat liver mitochondria exists as bilayer phase in the same 10–37 °C temperature range [114]. ^31^P-NMR spectra from aqueous dispersion of phospholipids of rat liver inner mitochondrial membrane were consistent with the bilayer arrangement of phospholipids which coexisted with a population of phospholipids that exhibited isotropic motional averaging typical for non-bilayer phospholipid structures [114]. A sample of intact rat liver mitochondria at 4 °C had ^31^P-NMR spectrum arising from phospholipids and small water-soluble phosphorus atom containing molecules like ADP and P_i_. The phospholipid component of this ^31^P-NMR spectrum reflected a bilayer lipid arrangement. At 37 °C, phospholipids in intact rat liver mitochondria were arranged in bilayer structures with a small population of phospholipids arranged in non-bilayer structures [114]. This report was probably the first indication of the existence of non-bilayer arranged phospholipids in mitochondrial membranes. About one decade later, a fraction of bovine liver mitochondrial proteolipids predominantly composed of F_0_ protein subunits and tightly bound cardiolipin was isolated. ^31^P-NMR spectrum taken from the aqueous fraction of these mitochondrial proteolipids had a line shape of somewhat broad symmetrical signal with a resonance peak at 6 ppm [96,115]. It was suggested that this ^31^P-NMR signal arises from non-bilayer packed cardiolipin molecules which are immobilized by binding to F_0_ subunits. This suggestion was later corroborated by the results of computer modeling study that linked the symmetrical ^31^P-NMR signal at 6 ppm to non-bilayer phospholipid structures with restricted molecular mobility [35]. One more decade later ^31^P-NMR spectrum taken from intact cauliflower mitochondria at 18 °C showed that phospholipids in cauliflower mitochondria are arranged in bilayer structures [86]. However, ^31^P-NMR spectra taken at 18 °C from cauliflower mitochondria treated with cardiotoxins CTI and CTII revealed two ‘non-bilayer’ signals superimposed on the bilayer signal of phospholipids [86]. One non-bilayer signal at 0 ppm originated from the phospholipids with rapid isotropic mobility and another non-bilayer signal at 6 ppm was assigned to the immobilized non-bilayer arranged phospholipids [86]. A series of powerful physical methods including ^1^H-NMR, ^2^H-NMR, EPR of spin probes in oriented phospholipid films, and differential scanning calorimetry have been used in addition to ^31^P-NMR spectroscopy to study structural aspects of the interaction of cardiotoxins with intact cauliflower mitochondria and with a model membrane with phospholipid composition similar to that of cauliflower mitochondria [86]. The results of this study allowed one to conclude that the non-bilayer signal at 6 ppm arises from a population of phospholipids found in intermembrane junctions between OMM and IMM [88]. These phospholipids, which predominantly included cardiolipins, had their molecular mobility restricted by their interactions with cardiotoxins localized in the center of intermembrane junctions [86]. 

### 7.3. Non-Bilayer Packed and Immobilized Cardiolipin Promotes ATP Synthase Activity

A research group led by one of the authors of this review paper has been working with mitochondria for over five decades and succeeded in developing a procedure for the high-quality mitochondrial preparation for the ^31^P-NMR studies [115]. This procedure was further modified by another author of this review paper in which mitochondria were purified from phosphates of non-phospholipid nature to simplify ^31^P-NMR spectra and facilitate the analyses of the molecular mobility and polymorphic transitions of phospholipids in mitochondria [34,35]. In this procedure, freshly isolated bovine heart mitochondria were osmotically shocked to release phosphates of non-phospholipid nature and then allowed to recover in isotonic solution, after which mitochondria were tested for viability and functional activity by measuring respiratory control index ratios and the amount of ATP produced by mitochondrial ATP synthase over time [34,35]. These tests showed that mitochondria were highly coupled and in a state of high structural and functional quality [34,35]. The intact mitochondria samples were free of phosphates of non-phospholipid nature and with native structure and high functional activity and allowed to conduct ^31^P-NMR experiments in which phospholipid concentration in mitochondria was estimated with high accuracy and in which ^31^P-NMR spectra and ATP production were measured in the same sample. This made it possible to relate polymorphic transitions of mitochondrial phospholipids directly to ATP synthase activity. This was important information for understanding the role of non-bilayer structures in production of mitochondrial ATP. ^31^P-NMR spectra of these mitochondria taken at 8 °C revealed only a lamellar phase—a bilayer packed phospholipid arrangement—and no non-bilayer structures [34,35,36], as shown in Figure 5, which was first published in [35]—coauthored by two authors of the present review paper. The ATP synthase activity of these mitochondria at 8 °C was very low (Figure 5) [34,35,36]. ^31^P-NMR spectra of the same mitochondria at 15 °C revealed two small non-bilayer signals, signal A at 0 ppm and signal B at 6 ppm, both superimposed on the ‘lamellar’ signal (Figure 5). Signal A at 0 ppm originates from non-bilayer phospholipids with high isotropic mobility which exchange with bilayer phospholipids [35,36]. Signal B at 6 ppm arises from non-bilayer phospholipids immobilized by interactions with mitochondrial proteins [34,35,36]. A percentage of non-bilayer arranged immobilized phospholipids responsible for a signal at 6 ppm was estimated from the integral intensity of the signal at 6 ppm remaining after applying a DANTE train of saturation pulses at the high-field peak of the lamellar spectrum (Figure 5). Once immobilized, non-bilayer phospholipids appeared at 15 °C, production of ATP increased three times (Figure 5). The percentage of immobilized non-bilayer phospholipids increased with the increase of temperature and plateaued at 40 °C. Up to 40 °C, the increase in the ATP production was directly proportional to the increase of the fraction of immobilized non-bilayer phospholipids. ATP production reached maximum at 40 °C and then declined at 43 °C (Figure 5). Increase in intensity of ^31^P-NMR signal A at 0 ppm has been observed throughout the temperature range between 8 °C and 43 °C while the ATP production declined at 43 °C (Figure 5). These data strongly suggested that the immobilized non-bilayer phospholipids, rather than the non-bilayer phospholipids responsible for ^31^P-NMR signal A at 0 ppm, are responsible for the increased production of ATP [34,35]. This suggestion was in agreement with findings of another experiment. The same bovine heart mitochondria were treated at 15 °C with cardiotoxin CTII (the total mitochondrial phospholipid to cardiotoxin molar ratio was 70), which increased the percentage of immobilized non-bilayer phospholipids 1.6 times and the ATP production was doubled in comparison to the untreated mitochondria. When the cardiotoxin-treated mitochondria were post-treated with the phospholipid-hydrolyzing enzyme PLA_2_, both the percentage of immobilized non-bilayer phospholipids and the ATP production decreased by a factor of 6 [34,35,36]. This is further strong experimental evidence demonstrating a direct link between the amount of non-bilayer immobilized phospholipids and the rate of ATP production [35].

### 7.4. DCCD-BPF of F_0_ Sector of ATP Synthase Tightly Binds Four Cardiolipin Molecules

The dicyclohexylcarbodiimide-binding protein (DCCD-BPF) is a part of the C8 subunit in bovine ATP synthase. It is a hydrophobic component of the F_0_ sector imbedded in the IMM which is directly involved in proton shuttling through the F_0_ sector [116]. It has been shown that DCCD-BPF and cardiotoxin CTII share similar membranotropic properties [34,35,36]. When DCCD-BPF and cardiotoxin CTII were reconstituted at pH 7.4 in two separate sets of multilamellar dispersions with phospholipid composition similar to that of the bovine heart mitochondria, both proteins triggered formation of immobilized non-bilayer phospholipids responsible for ^31^P-NMR signal at 6 ppm [34,35,36]. Decrease in pH from 7.4 to 3.0 in solutions of two sets of multilamellar dispersions, one reconstituted with DCCD-BPF and another with cardiotoxin CTII, induced an increase in immobilized non-bilayer phospholipids by 26.5% in the case of DCCD-BPF and by 31.7% in the case of cardiotoxin CTII [34,35]. This is an important observation, suggesting that increase in proton concertation in the intermembrane space promotes the formation of non-bilayer phospholipids immobilized by proteins in the IMM. Also, since it has been shown that cardiotoxin CTII is a cardiolipin-targeting protein [34,35,36,86], one can assume that DCCD-BPF also promotes formation of immobilized non-bilayer phospholipids in the IMM by directly interacting with cardiolipin.

Multilamellar dispersions reconstituted with DCCD-BPF or cardiotoxin CTII in the abovementioned study were made of phosphatidylcholine, cardiolipin, phosphatidylethanolamine, phosphatidic acid, and phosphatidylserine in molar ratios of 6.0, 2.5, 0.6, 0.5 and 0.4, respectively, to model the phospholipid composition of the IMM in bovine heart mitochondria [13,34,35,117]. To confirm that DCCD-BPF similarly to cardiotoxin CTII interacts strongly with cardiolipin but not with other phospholipids in the IMM to promote formation of immobilized non-bilayer phospholipid structures, ^31^P-NMR spectra of five phospholipid preparations were recorded [34,35]. Each phospholipid preparation was suspended in neutral buffer with 1% Triton-X-100, and each was made of one of the five phospholipids found in the bovine heart IMM, plus each preparation contained either DCCD-BPF or cardiotoxin CTII. The recording parameters for NMR spectrometer were set to detect phospholipids forming tiny micelles with Triton X-100 that have high isotropic dynamics and yield narrow ^31^P-NMR signals [34,35]. Phospholipids strongly bound to proteins and not forming tiny micelles with Triton X-100 have restricted mobility, which greatly broadens ^31^P-NMR signal and renders restricted phospholipids ‘invisible’ at the set recording parameters. By utilizing this method, it was demonstrated that one mole of either DCCD-BPF or cardiotoxin CTII strongly bound up to four moles of cardiolipin in solution of Triton X-100, but none of the other phospholipids found in the IMM [34,35]. Native PAGE electrophoregrams of DCCD-BPF and cardiotoxin CTII in a buffer with cardiolipin revealed that both DCCD-BPF and cardiotoxin CTII form complexes in which four proteins of either DCCD-BPF or cardiotoxin CTII tightly bind 16 cardiolipin molecules, but no other phospholipids found in the IMM formed tight complexes with DCCD-BPF or cardiotoxin CTII [34,35]. These experimental data strongly suggest that cardiotoxin CTII phenocopies the ability of DCCD-BPF to form lipid-protein oligomers by binding strongly cardiolipins [34,35]. Although cardiotoxin CTII and DCCD-BPF have different amino acid sequences, utilization of AutoDock Vina software for extensive docking analysis of the docked structures of cardiotoxin CTII and DCCD-BPF with cardiolipin and other phospholipids found in the IMM revealed that both proteins share similar physico-chemical properties on their molecular surfaces, which explain similar membranotropic behavior of both proteins when they interact with mitochondrial membranes [34,35]. However, cardiotoxins have more basic amino acid residues than DCCD-BPF which allow cardiotoxins at higher concentrations to induce permeabilization and subsequent lysis of mitochondrial membranes [34,35,36,86,87].

### 7.5. Non-Bilayer Packed Cardiolipin May Promote Lateral Transfer of Protons and Tight Clustering of ATP Synthases and Respirasomes 

The proteins of ETC and the ATP synthases are enriched in the cristae membranes [118] in which proton pumps are positioned in near proximity to the ATP synthases [119]. Proton pumps push H^+^ ions into intracristae space to promote transmembrane proton gradient needed for the catalytic activity of ATP synthases [32]. To test for the existence of transmembrane H^+^ gradient, the local pH values were recently determined in mitochondrial subcompartments of respiring yeast cells by applying pH-sensitive pHGFP fusion proteins [120]. Unexpectedly, no significant proton gradient across the inner membrane of respiring cells was determined [120]. This finding supports the kinetic coupling hypothesis in which closely clustered proteins of oxidative phosphorylation (OXPHOS) complexes are coupled through the lateral diffusion of protons on the surface of cristae membrane [121,122]. The excess of protons on the interface driving ATP synthesis was shown for the first time in the octane-water model [123]. A kinetic barrier on membrane surface does not allow H^+^ ions to detach immediately into the solution [124,125], which results in short-distance lateral transfer of H^+^ ions from proton pumps to ATP synthases within a tightly clustered OXPHOS system [126]. Indeed, lateral movement of protons on membrane surface that drives ATP synthesis has been observed in mitochondria and mitoplasts [127,128]. It has been proposed that cardiolipin with anionic phosphate groups, which attract H^+^ ions, may be responsible for the lateral transfer of protons on the surface of cristae membrane [32,33]. Attraction of H^+^ ions to cardiolipin’s phosphate groups strengthens the conical shape of cardiolipin, which triggers non-bilayer packing of cardiolipin; thus, in addition to lateral transfer of protons, may play a critical role in promoting a tight clustering of an OXPHOS system. A highly ordered linear oligomerization of tightly docked ATP synthases and respirasomes has been recently observed for the first time in the cristae of cardiac mitochondria by cryo-electron microscopy, as shown in Figure 6 [126]. 

It has been proposed that clustering of complexes in an OXPHOS system is mediated by the membrane raft-like structures containing cardiolipin, which are formed due to membrane surface curvature triggered by interaction of phospholipids with ATP synthases [126] and possibly with proteins of ETC. It is possible that non-bilayer packing of cardiolipin, which is induced by interaction with ATP synthases and other proteins of ETC, underlies the mechanism of membrane rafts formation. It has been also proposed that the clustering of the OXPHOS system is a dynamic process that may represent the functional state of IMM [126]. In the highly respiring mitochondria, the OXPHOS system is tightly clustered with virtually no proton leaks, while in the resting mitochondria, the OXPHOS system is loosely clustered featuring higher leaks of protons [126]. 

## 8. Conclusions 

The wealth of experimental data collected so far helps with putting together a conceptual model explaining how cardiolipin molecules work to facilitate the efficiency of the ATP production. The evidence presented in this review shows that cardiolipin strongly binds to DCCD-BPF [34,35,36] and probably to other subunits of F_0_ complex to keep the subunits tightly together. In addition, cardiolipin plays an essential role in the oligomerization of respiratory complexes into respirasomes [25]. Regulation of proper morphology of the inner mitochondrial membrane requires interaction of cardiolipin with OPA1, a protein mainly involved in fusion and tubulation of mitochondrial membranes [129,130,131]. Cardiolipin also interacts with cytochrome *c* and brings it closer to the ETC. This, in turn, accelerates the electron transfer from CIII to CIV [29], which is the rate-limiting step in electron transfer. Further, cardiolipin seems to be an important element in promoting the transmembrane proton gradient across the IMM, which could be accomplished through various mechanisms, including flip-flopping of protonated cardiolipin [31,32]. Cardiolipin is likely to be responsible for the lateral proton diffusion on membrane surface, which is a proton-shuttling across cristae membranes and/or sequestering protons by cardiolipin from the ETC with subsequent release of protons to the F_0_ subunit [32,33,106]. Cardiolipin also plays an essential role in ATP synthase dimerization [132], which is necessary for creation of cristae [133], and entrapment of protons in the cristae tips [30]. A crucial role of cardiolipin in the formation of supramolecular structures of ATP synthase dimers in *Drosophila* flight-muscle mitochondria was strongly validated when mutant *Drosophila* species with reduced cardiolipin levels showed significantly reduced density of ATP synthase complexes in flight-muscle mitochondria [23]. 

The role of non-bilayer structures in increasing the efficiency of ATP synthase complexes is becoming more and more evident. An increase in proton concentration on the surface of cristae membrane helps to neutralize phosphate groups of cardiolipin, which reduces the hydrate coat size around the polar head of cardiolipin to further increase its reverse wedge molecular shape [85]. This change in molecular shape mediates the transition from a bilayer to non-bilayer packing of phospholipids in cardiolipin containing membranes [34,35,84]. This may lead to the formation of small compartments with increased surface curvature in the apex of cristae with a bridge between cristae membranes as shown in Figure 7. Formation of such compartments in cristae was proposed by us in [34,36], and the figure depicting the structure of compartments in cristae was originally published by us in [35]. Such bridged compartments segregate the ATP synthase dimers and entrap higher concentration of protons along the membrane surface of the compartment [34,35,36]. Obviously, formation of bridged compartments is a transient process that takes place in highly respiring but not in resting mitochondria. Formation of bridges in cristae membranes has not been observed by cryo-electron microscopy. This might be explained by experimental findings that non-bilayer structures, some of which include bridges, were not observed by ^31^P-NMR spectroscopy at temperatures below 8 °C when mitochondria do not respire actively [35,114]. 

Figure 7 shows the compartment with increased surface curvature in the apex of cristae with cardiolipin molecules concentrated in places of compartment with the most membrane surface curvature. The compartment in Figure 7 includes four ATP synthases which make two dimers. Below the compartment there are two single ATP synthases on opposite membranes of cristae. Each of these ATP synthases may also dimerize with other ATP synthases not shown in Figure 7. This may occur when another bridge between cristae membranes is created to form a new compartment in which ATP synthases are brought close to each other to facilitate their dimerization. Formation of compartments also increases H^+^ concentration along the inner membrane surface in the compartments, which promotes stronger diffusion of protons to the F_0_ channel of the ATP synthase and increases the efficiency of ATP production [34,35]. It is possible that cardiolipin-targeting proteins, such as SS-peptides and cardiotoxins, promote formation of new compartments inside cristae by efficiently increasing the reverse wedge shape of cardiolipin. It is possible that cardiolipin-targeting proteins like SS-peptides and/or cardiotoxins are synthesized naturally by mitochondria or in cytoplasm to promote the oligomerization of ATP synthase complexes and the formation of supramolecular assemblies of ATP synthases recently discovered in Drosophila flight-muscle mitochondria [23] and the tightly clustered OXPHOS complexes discovered in cardiac mitochondria [126].

The concept of an increased reverse wedge shape of cardiolipin that promotes the formation of the new cristae and new non-bilayer compartments inside cristae to facilitate ATP production may open new pharmacological avenues for engineering novel pharmaceuticals designed to reverse declining mitochondrial bioenergetics and effectively treat cardiovascular diseases, including ischemia and heart failure, and other diseases associated with the decline in mitochondrial ATP production. 

## Figures and Tables

**Figure 1 cells-10-01721-f001:**
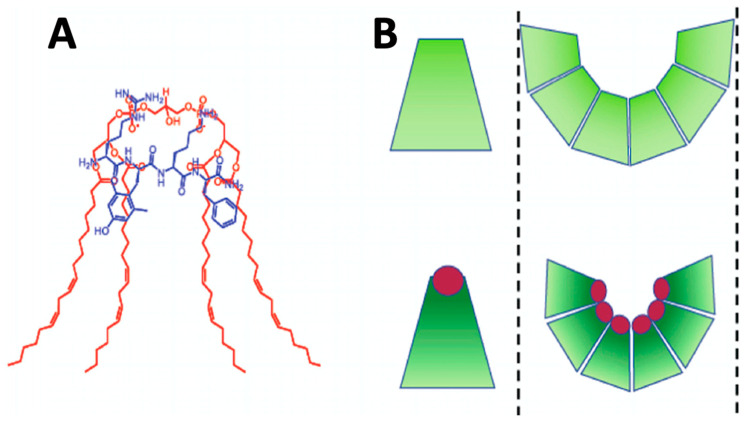
Interaction of cardiolipin (shown in red) with the Szeto–Schiller tetrapeptides (shown in blue) increases the conical shape of cardiolipin (**A**). Segregation of cardiolipin molecules with the increased conical shape in the inner monolayer of IMM exerts lateral pressure in the membrane, which induces negative curvature causing the inner monolayer of IMM to buckle and form cristae curvature (**B**). The images of the Szeto–Schiller tetrapeptides, the conically shaped cardiolipins and a cristae curvature are modified from [1] and reproduced with permission by Elsevier.

**Figure 2 cells-10-01721-f002:**
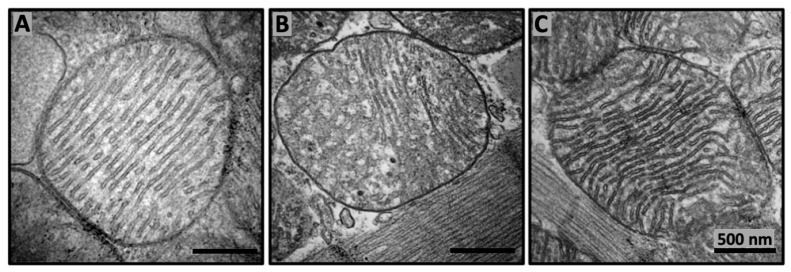
SS-31 repairs mitochondrial cristae architecture in aged mice. Representative transmission electron micrography images of murine cardiac mitochondria from 6 months old (**A**), 26 months old (**B**), and 26 months old treated with SS-31 (**C**). SS-31 was administered subcutaneously with daily dosage 1 mg/kg starting at age of 24 months. Transmission electron micrography images are modified from [[1] and reproduced with permission by Elsevier].

**Figure 3 cells-10-01721-f003:**
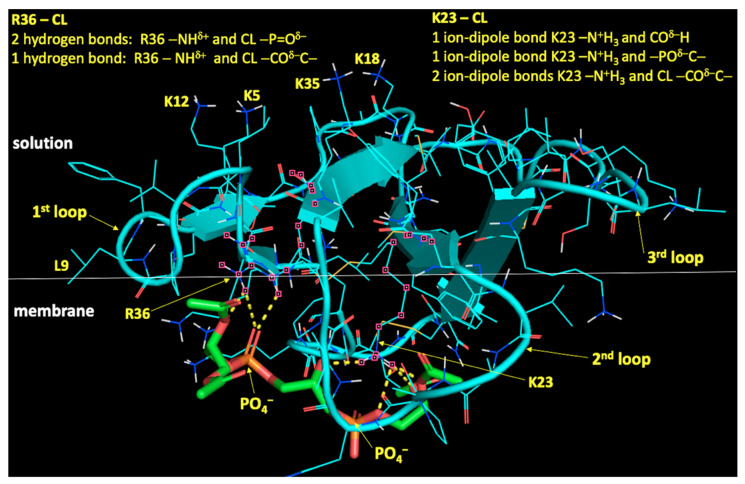
Pymol diagram of cardiotoxin CTII interaction with cardiolipin polar head shown in cartoon-lines (CTII) and cartoon-sticks (cardiolipin) representations modified from [37]. CTII binds to membrane surface in horizontal orientation with the three loop tips of CTII facing the viewer. Carbon atoms of cardiolipin are presented as green sticks. Atoms of amino acid residues interacting with cardiolipin are shown as pink squares. Intermolecular bonds are shown in yellow broken lines. Types of bonds are described in yellow letter sentences. In this particular orientation of cardiotoxin on a membrane surface, residues K5, K12, K18, and K35 (shown as cartoon lines) presumably bind anionic phospholipids of an adjacent membrane.

**Figure 4 cells-10-01721-f004:**
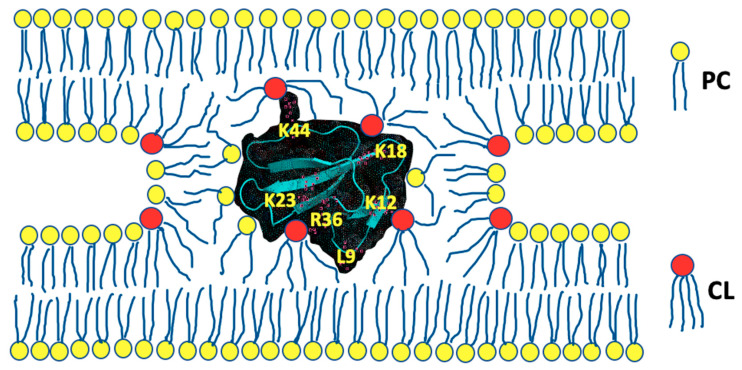
The diagram of intermembrane junction with a cardiotoxin CTII in its center; modified from [37]. Cardiotoxin CTII is given in a cartoon and dots representation. Amino acid residues R36 and L9 (predicted by molecular dynamics data) and K12 (predicted by AutoDock data) are key residues in the initial binding of cardiotoxin to phosphatidylcholine membrane containing 5 mol% cardiolipin. R36 and K12 bind to cardiolipin while L9 penetrates to the membrane’s hydrophobic region. K5, K35 (not shown in this figure as they are on another side of cardiotoxin), K18 and K44 presumably attract anionic phospholipids of a neighboring lysosome. K23 and some other residues, which are not shown in this figure, establish short-range intermolecular bonds with polar groups of cardiolipin and PC to support further embedding of cardiotoxin into a membrane.

**Figure 5 cells-10-01721-f005:**
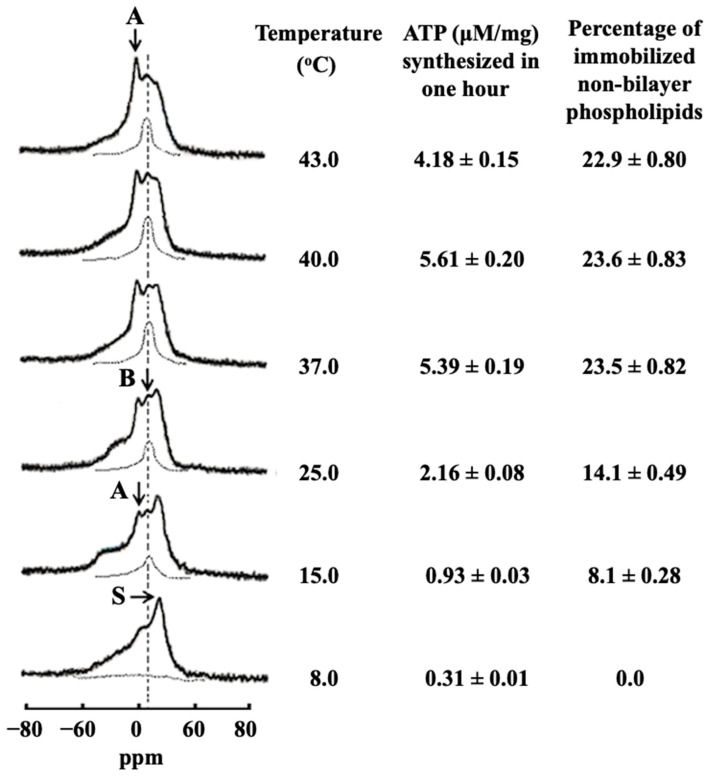
Temperature triggered formation of non-bilayer structures increases the ATP production in mitochondria; this figure is reproduced from [35]) with permission by Elsevier. ^31^P-NMR spectra of mitochondria recorded at different temperatures in the presence of 1.5 mM succinate. The phospholipid concentration for each mitochondrial sample was estimated to be 6.3 × 10^−2^ M as assessed by normalizing the integral intensity of the ^31^P-NMR signals from the mitochondrial samples to the integrated intensity of the ^31^P-NMR signals measured from multilamellar liposomes of known phospholipid concentrations. Hatched lines are saturation spectra observed after applying a DANTE train of saturation pulses at the high-field peak of the lamellar spectrum (see arrow with letter S). Position of the hatched line signals in the saturation spectra coincides with the position of ^31^P-NMR signal B at 6 ppm arising from non-bilayer immobilized phospholipids. The amounts of these non-bilayer immobilized phospholipids were estimated by calculating the areas under the hatched ^31^P-NMR lines below the signal B after a DANTE train of saturation pulses. The ^31^P-NMR signal A at 0 ppm originates from non-bilayer phospholipids with rapid isotropic motion. ATP levels expressed as μmol ATP synthesized per mg of mitochondrial proteins in 60 min, was monitored by taking measurements on aliquots from the ^31^P-NMR sample tubes. Variance in measurements between different samples was ±4%.

**Figure 6 cells-10-01721-f006:**
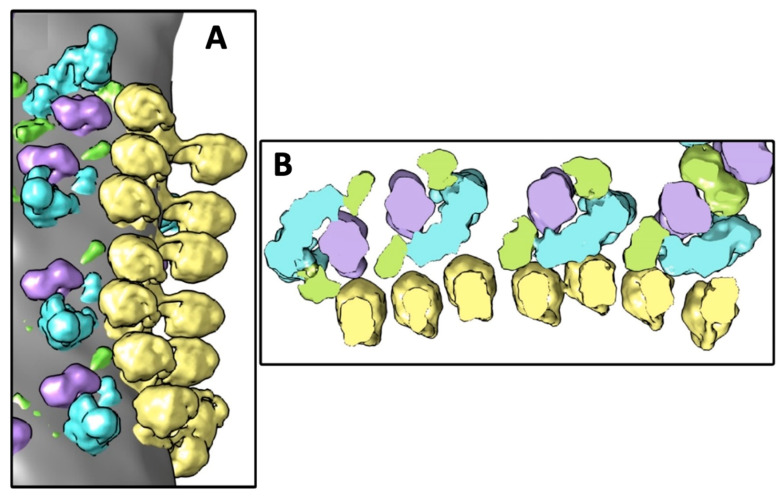
(**A**)—Surface rendering of the selected area showing the oligomeric linear structure consisting of tightly docked ATP synthases and respirasomes. Elongated transmembrane parts of complexes I are oriented approximately parallel to the row of ATP synthases. (**B**)—Slice of the density map, view from the intermembrane space. ATP synthase, complex I, complex III dimer, and complex IV are shown in yellow, blue, purple, and green colors respectively. Images in A and B are modified from [126].

**Figure 7 cells-10-01721-f007:**
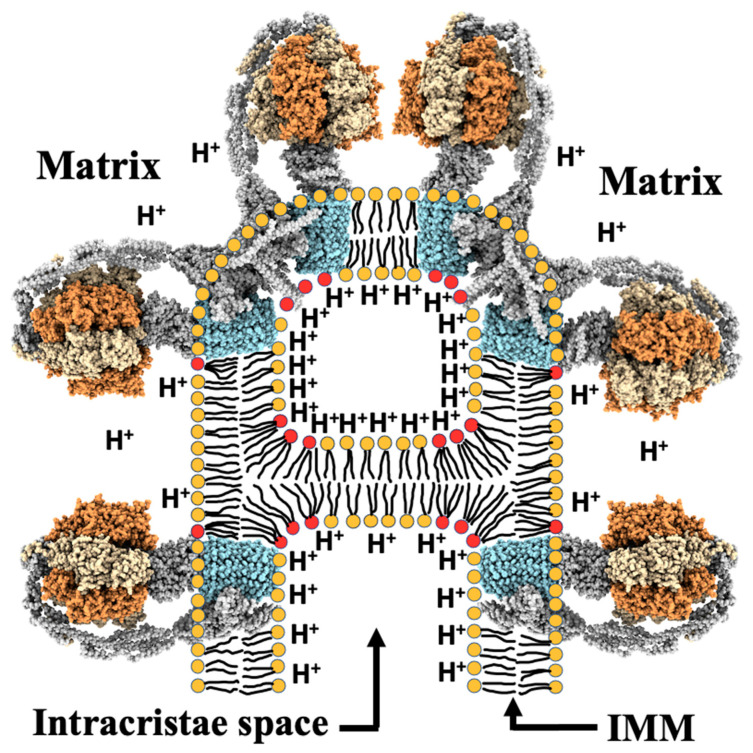
Schematic representation of a compartment with increased surface curvature in the apex of cristae is modified from [35,134] with permission by Elsevier. This is formed when cardiolipin changes bilayer packing to non-bilayer packing and is mostly found in places of compartment membrane with the high surface curvature. Only ATP synthases are shown, no other complexes of the OXPHOS system are displayed. Increased concentration of protons is shown along the inner surface of the compartment. The structure of the dimeric F_0_ complex from *Saccharomyces cerevisiae* resolved at 3.6 angstroms (PDB # 6b8h) [116] shows the same angle between the dimers as found in heart mitochondria. PDB # 6b8h coordinates were used by Dr. S. V. Nesterov of Moscow Institute of Physics and Technology for reconstruction with the UCSF ChimeraX program [135,136] the atomic structure of dimeric and monomeric forms of ATP synthases shown in this figure. The α- and β-subunits of catalytic domain are shown in yellow and orange, respectively; the *c* ring is given in blue. All other subunits are presented in grey. Phospholipids with red polar heads represent cardiolipin. Phospholipids with yellow polar heads represent other phospholipids with two acyl chains found in the IMM. For better visibility, acyl chains of phospholipids are not drawn on top of ATP synthase subunits imbedded in cristae membrane.

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
