# Peer review of "Cardiolipin, Non-Bilayer Structures and Mitochondrial Bioenergetics: Relevance to Cardiovascular Disease"

_cells, 2021, doi:10.3390/cells10071721_

Round 1
Reviewer 1 Report
This review by the Dr. Gasanoff and colleagues encompasses a subject of great interest to a broad readership, namely cardiolipin role in mitochondrial bioenergetics with respect to cardiovascular diseases.
This is a very informative, interesting and thorough review. Overall, the manuscript is very well written and was a real pleasure to read.
Author Response
Response to Reviewer 1 Report
We are deeply grateful to the reviewer for his/her detailed attention to the content of our manuscript and for his/her positive opinion about this review paper.
Reviewer 2 Report
This manuscript is a review article aimed to provide information about cardiolipin, non-bilayer structures and mitochondrial bioenergetics. The authors outlined the link between changes in mitochondrial cardiolipin concentration and changes in mitochondrial bioenergetics. They also discussed the effects of cardiolipin concentration changes in the IMM curvature and surface area, cristae density and architecture, efficiency of electron transport chain (ETC), interaction of ETC proteins, oligomerization of respiratory complexes, and mitochondrial ATP production. The cardiolipin and cardiovascular diseases were discussed. Non-bilayer structures and their correlation with cardiolipin were elaborated in the manuscript. The manuscript collected enough information and organized the structure well. It should be interesting to researchers in the field.
Concerns
- Figure 5 is highly similar to Figure 1 of BBA - Biomembranes 1860 (2018) 586–599. Figure 7 is also similar to figure 7 in the Biomembranes article. It is highly recommended that authors clearly indicate the origins in the main text.
- The manuscript contains many unconventional scientific wordings. Professional English editing is highly recommended.
- As a reviewer article, this manuscript emphasizes the link between cardiolipin and cardiovascular diseases. It will be nice to see the authors add more section to discuss the molecular mechanisms of cardiolipin and the pathogenesis of cardiovascular diseases.
Author Response
We are deeply grateful to the reviewer for his/her detailed attention to the content of our manuscript. We believe that the reviewer’s valuable and constructive comments are made to improve the quality of our manuscript. Below are our responses to the points of concerns raised by this reviewer.
Point 1: Figure 5 is highly similar to Figure 1 of BBA - Biomembranes 1860 (2018) 586–599. Figure 7 is also similar to figure 7 in the Biomembranes article. It is highly recommended that authors clearly indicate the origins in the main text.
Response 1: As per the reviewer’s recommendation, in addition to the references to Figures 1 and 7 in our original article in BBA - Biomembranes 1860 (2018) 586–599, we have clearly indicated the origins of Figures 5 and 7 given in our present manuscript in the main text of our present manuscript in lines 586-588 and 762-764 (shaded yellow).
Point 2: The manuscript contains many unconventional scientific wordings. Professional English editing is highly recommended.
Response 2: To address the reviewer’s concern about unconventional scientific wordings and his/her recommendation for the professional English editing, we have thoroughly revised our manuscript and corrected grammar, punctuation, and typos. We also edited a few scientific wordings to improve the flow of English (e.g. in lines 20, 24, 25, 47, 51, 86, 111, 119, 126-129, 137, 154, 156, 158, 160, 178-180, 194, 195, 197, 198, 200, 218, 219, 227, 228, 262, 276, 313, 316-118, 320, 322, 361, 363, 366, 370, 385, 404, 408, 425, 427, 450, 458, 469, 523, 524, 535, 536, 538, 543, 551-553, 555-557, 561-562, 575, 582, 589-591, 593, 596, 599-601, 604, 605, 607, 609-614, 620, 621, 625-631, 636, 656, 661, 670, 674, 675, 677, 689, 690, 692, 695, 702, 708, 709, 726, 727, 733, 736, 740, 741, 745, 755, 759, 760, 764, 767, 768, 775, 776, 781, 782 794, 795, 800, 801 and 808 (yellow labels). Although we believe that English language in our manuscript is in a proper order now, we will gladly accept any further English editing by the Cells editors should they decide to further edit English language in our manuscript.
Point 3: As a reviewer article, this manuscript emphasizes the link between cardiolipin and cardiovascular diseases. It will be nice to see the authors add more section to discuss the molecular mechanisms of cardiolipin and the pathogenesis of cardiovascular diseases.
Response 3: We do share the reviewer’s interest to see more information related to the molecular mechanisms of cardiolipin and the pathogenesis of cardiovascular diseases. From the current knowledge on this issue, we know that the pathogenesis of cardiovascular diseases is linked to the decline in cardiolipin concentration, which leads to abnormal structures and inefficient remodeling of cristae architecture, a decrease in the formation of respirasomes and in the efficiency of ETC, MICOS, OPA1, ATP synthases, and an overall decline in the production of ATP - all these are now described in our revised manuscript. Triggered by the reviewer’s comment, we have added a subsection (ll. 462-520) devoted to our understanding about how the high propensity of cardiolipin in forming non-bilayer structures in the IMM may support the key processes behind the efficient remodeling of cristae architecture and how this may strikingly outline one more aspect related to the pathogenesis linked to the abnormal decline in the mitochondrial cardiolipin levels that leads to cardiovascular diseases.
Reviewer 3 Report
- General comment
In this review, it was well discussed about the current knowledge on the roles that cardiolipin, predominantly found in the inner mitochondrial membrane, play in mitochondrial bioenergetics. The review represents a great advance in the understanding the significance of cardiolipin for mitochondrial ATP synthase dimerization and ATP production, mitochondrial dysfunction leading to various diseases including cardiovascular disease and so on.
- Major revision
1) Line 77~79 and Figure 1
It is strongly recommended to show the chemical structure of cardiolipin, similarly to Fig. 2b (and Fig. 2a) in Ref. 1), in addition to its conical molecular shape of cardiolipin in Fig. 1.
2) Fig. 6
As it is a little bit difficult to understand the oligomeric structure of tightly docked ATP synthases and respirasomes, it is strongly recommended to add Figure 2D of Ref. 109), in addition to Fig. 6.
- Minor revision
- Line 265~267: It is recommended to show the amino acid sequence of SS peptide, shown in the legend of Fig. 1 of Ref. 1), as in the followings.
- SS-20 (H-Phe-D-Arg-Phe-Lys-NH2) and SS-31 (H-D-Arg-Dmt-Lys-Phe-NH2). Dmt= dimethyltyrosine.
- As it is difficult to understand the following sentence, it is necessary to check it.
- Line 676~680: Each of these ATP synthases may also dimerize should another compartment be formed below the compartment formed in the apex of cristae.
Author Response
- Major revision
Point 1: Line 77~79 and Figure 1
It is strongly recommended to show the chemical structure of cardiolipin, similarly to Fig. 2b (and Fig. 2a) in Ref. 1), in addition to its conical molecular shape of cardiolipin in Fig. 1.
Response 1: As per the reviewer’s recommendation, we have added to Fig. 1 the skeletal structure of cardiolipin interacting with the SS tetrapeptide (as in Fig 2b in Reference 1) and made appropriate edits in the Fig. 1 legend (ll. 104-108, yellow labelled). We have also added comments to Fig. 1 in the main text (ll. 94-96, yellow labelled). We felt that Fig. 2a in Reference 1 is somewhat redundant and we didn’t add it to Fig. 1, which allowed better visibility of the skeletal structure of cardiolipin interacting with the SS tetrapeptide.
Point 2: Fig. 6
As it is a little bit difficult to understand the oligomeric structure of tightly docked ATP synthases and respirasomes, it is strongly recommended to add Figure 2D of Ref. 109), in addition to Fig. 6.
Response 2: As per the reviewer’s recommendation, we have added to Fig. 6 Figure 2D of Reference 109, which is now Reference 126. We have also added a sentence in the legend to Fig. 6 (l. 717, yellow labelled), relevant to the amended Fig. 6.
- Minor revision
Point 3: Line 265~267: It is recommended to show the amino acid sequence of SS peptide, shown in the legend of Fig. 1 of Ref. 1), as in the followings.
SS-20 (H-Phe-D-Arg-Phe-Lys-NH2) and SS-31 (H-D-Arg-Dmt-Lys-Phe-NH2). Dmt= dimethyltyrosine.
Response 3: As per the reviewer’s recommendation, we have shown the amino acid sequences of SS peptides (ll. 295 and 297, yellow labelled).
Point 4: As it is difficult to understand the following sentence, it is necessary to check it:
Line 676~680: Each of these ATP synthases may also dimerize should another compartment be formed below the compartment formed in the apex of cristae.
Response 4: As per the reviewer’s recommendation, we clarified the sentence (now, ll. 790-793, yellow labelled); we trust that the revised version conveys better the possible dimerization of the two single ATP synthases shown in Fig. 7; which may occur when another bridge between cristae membranes is formed to make a new compartment in which other ATP synthases, not shown in Fig 7, are brought close to the two single ATP synthases displayed.
Reviewer 4 Report
The review article “Cardiolipin, Non-bilayer Structures and Mitochondrial Bioenergetics: Relevance to Cardiovascular Disease” is dedicated to analysis of the link between changes in mitochondrial cardiolipin concentration and changes in mitochondrial bioenergetics that includes changes in the IMM curvature and surface area, cristae density and architecture, efficiency of electron transport chain (ETC), interaction of ETC proteins, oligomerization of respiratory complexes, and mitochondrial ATP production. The authors of the article suppose that the relationship between cardiolipin decline in IMM and mitochondrial dysfunction leading to various diseases including cardiovascular disease.
The article is well written.
The study has a good design.
The article is logically divided into sections and subsections.
In the article there are no grammatical and stylistic errors.
There is a table and many figures of good quality presented in the article.
The references cited are relevant and adequate.
A large number of scientific literature sources were analyzed.
In my opinion, this review paper can be recommended for publication after minor revision.
It is recommended to separate sections into subsections to make the article easier for understanding.
It is recommended to include a list of abbreviations, used in the article.
It is recommended to add articles of 2020-2021 to the list of references.
Author Response
Response to Reviewer 4 Comments
We are deeply grateful to the reviewer for his/her detailed attention to the content of our manuscript. We believe that the reviewer’s valuable and constructive comments helped us to improve the quality of our work. Below are our responses to the comments raised by this reviewer.
Point 1: It is recommended to separate sections into subsections to make the article easier for understanding.
Response 1: As per the reviewer’s recommendation, we separated sections into subsections to make it easier for the readers to understand the content of our manuscript.
Point 2: It is recommended to include a list of abbreviations, used in the article.
Response 2: As per the reviewer’s recommendation, we have included a list of abbreviations (ll. 37-42).
Point 3: It is recommended to add articles of 2020-2021 to the list of references.
Response 3: As per the reviewer’s recommendation, we have added a new subsection (ll. 462-520) containing 12 new references from 2020-2021. This new subsection describes the role of cardiolipin propensity to form non-bilayer structures in a process of remodeling the cristae architecture. The 12 new references are to be found: ll. 1022-1047. Originally, our manuscript listed 8 references of 2020-2021 in lines 898, 974, 976, 1059, 1079, 1082, 1086 and 1088. Now with addition of 12 new references of 2020-2021, our manuscript lists 20 references of 2020-2021. We are grateful for this remark of Reviewer 4, which prompted us to bring more recently published papers to the attention of readers of Cells.
Round 2
Reviewer 2 Report
If the editor checked there is no copyright issue with the figure 5 and
figure 7. I have no further concerns about the manuscript.